# Transfer learning from synthetic data reduces need for labels to segment brain vasculature and neural pathways in 3D

**Johannes C. Paetzold**[*1,†]**, Oliver Schoppe**[*1]**, Rami Al-Maskari**[1]**, Giles Tetteh**[1]**, Velizar Efremov**[1]**, Mihail I. Todorov**[2]**, Ruiyao Cai**[2]**, Hongcheng Mai**[2]**, Zhouyi Rong**[2]**, Ali Ertuerk**[2]**, Bjoern H. Menze**[1,†]

[1] *TranslaTUM and Department of Computer Science, Technical University of Munich, Germany*

[2] *Institute for Stroke and Dementia Research, Ludwig Maximilian University of Munich, Germany*

[†] *Correspondence to johannes.paetzold@tum.de and bjoern.menze@tum.de*

**Editors:** Under Review for MIDL 2019

## Abstract

Novel microscopic techniques yield high-resolution volumetric scans of complex anatomical structures such as the blood vasculature or the nervous system. Here, we show how transfer learning and synthetic data generation can be used to train deep neural networks to segment these structures successfully in the absence of or with very limited training data.

**Keywords:** Deep learning, transfer learning, synthetic data, vasculature, neural pathways.

## 1. Introduction

Recent advances in tissue-clearing (Ertürk et al., 2012; Chung and Deisseroth, 2013) combined with 3D light-sheet microscopy (*3D LSM*) overcome previous imaging limitations: they enable volumetric acquisition at cellular resolution of entire organisms (Cai et al., 2018; Pan et al., 2019; Stefaniuk et al., 2016; Mano et al., 2018). This yields unprecedented insight into the micro-anatomy at the macro-scale, e.g., to study highly connected structures like the brain vasculature or the peripheral nervous system. Differences in these structures have been associated with a wide range of disorders (Joutel et al., 2010; Li et al., 2010). Thus, segmentation and characterization of these anatomical structures is crucial to study causes and effects of such pathologies. However, manual segmentation of complex structures is very time-consuming, especially in high-resolution volumetric scans. While this motivates the need for deep learning it also implies a high cost of labeling. Here, we substantially reduce the need for manually labeled training data using transfer learning, an approach gaining attention (Van Opbroek et al., 2015; Khan et al., 2019). In short, we show that training deep networks on synthetic data is already sufficient to learn the basic underlying task across different anatomical structures, species, and imaging modalities.

## 2. Methods

Here, we present results from three widely different applications: human brain vessels (MRI), mouse brain vessels and the mouse peripheral nervous system (both *3D LSM*).

---

[*] Contributed equally

The same network was trained either on a small labeled set from the respective application ("real data"), on synthetically generated data, or on a combination of both. The synthetic data used is identical for all three applications. We chose DeepVesselNet as our architecture; the schedule for pre-training on synthetic data and refinement on real data match the methods of (Tetteh et al., 2018). The methods for generation of synthetic training data is described in (Schneider et al., 2012). MRI scans from human brain vasculature are taken from (Tetteh et al., 2018) (voxel size: $300\mu$m x $300\mu$m x $600\mu$m). Volumetric scans of the brain vasculature (voxel size: $(3\mu\text{m})^3$) and the peripheral nervous system (voxel size: $(10\mu\text{m})^3$) were obtained using DISCO tissue clearing and fluorescent light-sheet microscopy as described in (Cai et al., 2018). Representative 2D cross-sections of the synthetic data and segmentations of all three applications are shown in Figure 1.

## 3. Results

**Transfer learning from synthetic data (Table 1, Part 1).** For segmenting the human vasculature from MRI scans, training the net on the synthetic data alone yields very good results, 81% in F1-score (note: the synthetic data set had been designed for this application). Training on the real data for this application yields a higher F1-score of 86%. The best result (87%), however, is achieved by a combination of both: pre-training on synthetic data and fine-tuning on real data. Interestingly, the network also converges about 50% faster in this case (data not shown). Motivated by this observation, we repeated this experiment for *3D LSM* scans of the mouse brain vasculature. Again, the same pattern can be observed and the combination of synthetic with real data (F1-score of 76%) outperforms synthetic data (71%) or real data alone (73%). Taking the approach yet further, we applied the approach to *3D LSM* full body scans of the peripheral nervous system of a mouse. While training on synthetic data alone was not very successful (16%) as compared to real data (49%), the gain from combining both was almost completely additive (64%).

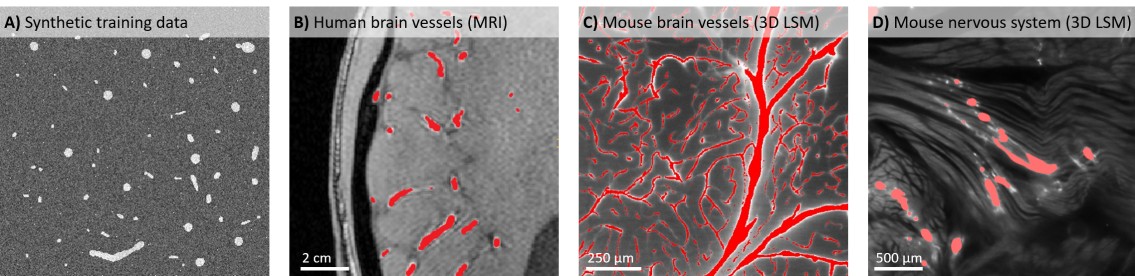

Figure 1: A) Synthetic training data was designed to resemble vasculature of human brain in MRI scans. B-D) Predicted segmentations of 3 different applications: MRI scans of human brain vasculature (B), *3D LSM* of mouse brain vasculature (C), and peripheral nervous system (D; shown here: innervated muscle fibres)

**Transfer learning across domains (Table 1, Part 2).** Here, we trained the network on a combination of synthetic data and the real data from a given application and then predicted on data from another application. When predicting on human vasculatures from

MRI scans, the refinement step on real data from another application after pre-training on synthetic data leads to worse results (left column: 43% and 36%) compared to training on synthetic data alone (81%, see Part 1). However, when training the model on synthetic data and real data of human vessels in MRI scans (first row of Part 2), the performance on *3D LSM* scans of mouse brain vessels (72%) or the mouse peripheral nervous system (49%) is about as good as when trained on the respective real data alone. Also, while the domain transfer from mouse vasculature to mouse nervous system only yields mediocre results (35%), it works well the other way around: refining a model trained on synthetic data with real data from the nervous system to segment brain vessels almost works as well (75%) as if it had been refined on data within the same domain (76%, see Part 1).

| | Training set | Application #1 Human brain Vasculature MRI | Application #2 Mouse brain Vasculature 3D microscopy | Application #3 Mouse body Neural pathways 3D microscopy |
|---|---|---|---|---|
| **Part 1)** **Tranfer learning from synthetic data within domain** | Synthetic data only | 81% | 71% | 16% |
| | Real data only | 86% | 73% | 49% |
| | Synthetic + real data | **87%** | **76%** | **64%** |
| **Part 2)** **Transfer learning across application domains** | Synthetic + human vessel MRI data | *n/a* | 72% | 49% |
| | Synthetic + mouse vessel microscopy data | 43% | *n/a* | 35% |
| | Synthetic + mouse neuron microscopy data | 36% | 75% | *n/a* |

Table 1: Quality of predicted segmentations (F1-score) for 3 different applications

## 4. Discussion

Our results demonstrate how pre-training on synthetically generated data can accelerate model convergence and boost the overall segmentation performance. For a given desired performance, this thus means a reduced need for manually labeled training data, which is very expensive for complex structures in 3D scans. Importantly, a single synthetic data set that was originally designed to represent human vessels also works well for applications from different species, anatomical structures, and imaging modalities. This suggests that the features learned from the synthetic data are of general use for the abstract segmentation tasks, highlighting the generalizability of the approach. Thus, the expensively labeled data for a given application does not have to be used to learn a basic task but rather can be preserved for refining the pre-trained model to the specifics of the application (such as contrast, noise, background structures). Interestingly, this approach may also be of use in cases where no training data is available at all. For instance, we could show that a model trained on synthetic data and real data from another application can match the performance of a model trained from scratch on real data from the application of interest. Together, these results highlight the importance of transfer learning towards the goal of resolving a key bottleneck in adoption of deep learning: the high cost of data annotation.

## Acknowledgments

This work was supported the German Federal Ministry of Education and Research via the Software Campus initiative (to O.S.), the Vascular Dementia Research Foundation, Synergy Excellence Cluster Munich (SyNergy), ERA-Net Neuron (01EW1501A to A.E.). Furthermore, NVIDIA supported this work with a Titan XP GPU via the GPU Grant Program.

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
