# OpenReview forum: "Transfer learning from synthetic data reduces need for labels to segment brain vasculature and neural pathways in 3D"
_MIDL.io/2019/Conference/Abstract — MIDL Abstract 2019_

### Official Review · AnonReviewer1 · 2019-04-30
**Demonstration of transfer learning using synthetic data**

**Rating:** 2
**Confidence:** 2

**Review:**


Summary:

In this work, transfer learning is demonstrated using synthetic data and cross domain training data for the segmentation of vasculature in human and mice brain data. Two cases of transfer learning of learning across domains : synthetic and real, and human and mice brains, is evaluated. The results show that pre-training using synthetic data can improve F1 scores of the predicted segmentations.

Comments:

+ Focus on reducing  dependency on high quality labels by using transfer learning is reasonable

- The results across domains (humans and mice) is the more natural setting for transfer learning and this shows little improvement or in some cases a degradation in performance. It is a curious thing that the pretraining across domains hampers the performance to this extent; this fact warrants a discussion.

Other remarks:

The small improvements in performance using synthetic data (within domain for brains) is interesting. However, it would be further interesting to attempt data augmentation strategies that are more commonly utilised in deep learning to give better insight into if the improvement was due to the quality of the synthetic data or if the models were trained with insufficient training data.

---

### Official Review · AnonReviewer2 · 2019-04-30
**Interesting application of transfer learning to novel dataset**

**Rating:** 3
**Confidence:** 2

**Review:**

The authors present an application of transfer learning for vessel segmentation to tackle the issue of labour-intensive dense data labeling. They demonstrate the utility of pre-training on synthetic data as well as on different datasets - namely human brain vessels from MRI, and mouse brain  vessels and mouse peripheral nervous system, both 3D light-sheet microscopy (LSM). The motivation and methods are clearly presented.

As an applications paper, the authors demonstrate clear results on an interesting dataset, showing that the task of vessel segmentation can be largely learned from synthetic data. While the referenced paper (Tetteh et al., 2018) (submitted for journal review) demonstrates the clear advantage of pre-training on the chosen synthetic dataset for application to brain MRA, the presented abstract further demonstrates the advantages of such an approach on novel 3D LSM data. Additionally, the authors demonstrate that training on synthetic data+alternate source of real data performs almost as well as training only on real data of a given source, improving over training on synthetic data alone. This is an interesting finding.

The authors should clarify what they mean be 'transfer learning' - typically this involves fine-tuning on a target dataset after pre-training on a different dataset. The way it is presented however suggests that no fine-tuning is performed except for where 'synthetic + real data' is indicated in Part (1) of Table 1, i.e. Part (2) of Table 1 involves no fine-tuning on the target dataset - is this correct?

While the 3D MRI data and its pre-processing is explained in (Tetteh et al.,2018), more detail on the 3D-LSM dataset would have been helpful. The result of pretraining on synthetic+mouse vessel LSM data and evaluated on mouse neuron LSM data performs worse than pretraining on snythetic+human vessel MRI and evaluating on mouse neuron LSM. One might expect image characteristics to be more similar for the two LSM datasets and thus the latter to perform better, but it seems likely the latter was a smaller dataset or preprocessed quite differently to the brain MRI dataset which may have yielded this performance.

---

### Decision · Program_Chairs · 2019-05-06
**Acceptance Decision**

Accept